# Endothelin signalling mediates experience-dependent myelination in the CNS

**Matthew Swire[1,2]\*, Yuri Kotelevtsev[3], David J Webb[4], David A Lyons[2], Charles ffrench-Constant[1]**

[1]MRC Centre for Regenerative Medicine, MS Society Edinburgh Centre, University of Edinburgh, Edinburgh, United Kingdom; [2]Centre for Discovery Brain Sciences, University of Edinburgh, Edinburgh, United Kingdom; [3]Centre for Neurobiology and Brain Restoration, Skoltech Institute for Science and Technology, Moscow, Russian Federation; [4]British Heart Foundation Centre of Research Excellence, Centre of Cardiovascular Science, Queen's Medical Research Institute, University of Edinburgh, Edinburgh, United Kingdom

**Abstract** Experience and changes in neuronal activity can alter CNS myelination, but the signalling pathways responsible remain poorly understood. Here we define a pathway in which endothelin, signalling through the G protein-coupled receptor endothelin receptor B and PKC epsilon, regulates the number of myelin sheaths formed by individual oligodendrocytes in mouse and zebrafish. We show that this phenotype is also observed in the prefrontal cortex of mice following social isolation, and is associated with reduced expression of vascular endothelin. Additionally, we show that increasing endothelin signalling rescues this myelination defect caused by social isolation. Together, these results indicate that the vasculature responds to changes in neuronal activity associated with experience by regulating endothelin levels, which in turn affect the myelinating capacity of oligodendrocytes. This pathway may be employed to couple the metabolic support function of myelin to activity-dependent demand and also represents a novel mechanism for adaptive myelination.

DOI: https://doi.org/10.7554/eLife.49493.001

\*For correspondence:
mswire@ed.ac.uk

**Competing interests:** The authors declare that no competing interests exist.

## Introduction

There is increasing evidence that experience regulates CNS myelination. For example, social interactions, sensory stimulation and several forms of learning have been shown to alter white matter and myelin structure in both humans and animal models (*Scholz et al., 2009*; *Makinodan et al., 2012*; *Liu et al., 2012*; *Sampaio-Baptista et al., 2013*; *McKenzie et al., 2014*; *Etxeberria et al., 2016*; *Xiao et al., 2016*; *Hughes et al., 2018*). At the level of individual neurons and axons, increasing the level of activity by using optogenetics or chemogenetics enhances the generation of myelin-forming oligodendrocytes and increases the amount of myelin they form (*Gibson et al., 2014*; *Mitew et al., 2018*), whilst preventing synaptic vesicular release from axons reduces myelin formation (*Hines et al., 2015*; *Mensch et al., 2015*; *Koudelka et al., 2016*). Together, these findings have led to a new concept of CNS plasticity - adaptive myelination. This concept posits that changes in neuronal activity in response to experience of the extrinsic environment lead to local changes in myelination. Such changes could in turn contribute to the alterations in conduction that underpin CNS neural circuit plasticity (*Sampaio-Baptista and Johansen-Berg, 2017*; *Foster et al., 2019*; *Suminaite et al., 2019*).

Exploring this important concept requires that we understand the mechanisms that can link changes in neuronal activity to the regulation of myelination. In addition to the communication between axons of active neurons and the oligodendrocytes that myelinate them, neuronal activity might also affect myelination indirectly in the local area through signals from other glial cells or from vascular cells that can respond to dynamic changes in neuronal activity. Such indirect signalling has been implied by in vitro studies, wherein increased levels of LIF secreted by cultured astrocytes in the presence of neuronal activity enhances myelination (*Ishibashi et al., 2006*). However, whether the vasculature might relay information about neuronal activity and in turn influence myelination by oligodendrocytes in vivo is not known.

Here we set out to test the hypothesis that the vasoactive peptide endothelin (EDN) enables blood vessels to play such a role in indirectly linking activity and myelination. This hypothesis is based on two sets of prior data. First, EDN expression by endothelial cells increases with enhanced blood flow (*Yanagisawa et al., 1988*; *Dancu et al., 2004*; *Walshe et al., 2005*; *Pandit et al., 2015*), which occurs in response to increased CNS activity. Second, the EDN G-protein coupled receptor (GPCR) endothelin receptor B (EDNRB) enhances myelination in slice cultures (*Yuen et al., 2013*). Here we show that EDN is expressed by CNS vascular cells, and that this expression lessens in the medial prefrontal cortex following social isolation, which we confirm also leads to impaired cortical myelination. Correspondingly, we show, by manipulating EDNRB signalling in rodent and zebrafish models, that reduced EDN signalling decreases the number of myelin sheaths formed by individual oligodendrocytes in vivo. Finally, we rescue the reduction in myelination associated with environmental social deprivation by intranasal administration of an EDNRB agonist to activate EDN signalling within the CNS. Together our data indicate that the vasculature responds to environmental signals associated with changes in neuronal activity and can, in turn, affect the myelinating capacity of oligodendrocytes in vivo. In this way, we propose a novel mode by which active neurons may regulate oligodendrocyte behaviour and provide a mechanism for adaptive myelination.

## Results

### The levels of EDN expression in blood vessels are reduced following social isolation

If EDN signalling from CNS blood vessels provides a link between circuit function and myelination, then an initial prediction would be that EDN expression by these vessels will be responsive to extrinsic environmental changes that alter neuronal activity and myelination. To test this prediction, we used a model in which the extrinsic environment has been shown to regulate CNS myelination: social isolation (*Liu et al., 2012*; *Makinodan et al., 2012*).

Prior work has established that social isolation in mice during a critical period comprising 2 weeks after weaning reduces both the excitability of specific subtypes of pyramidal neurons of the medial prefrontal cortex (mPFC) and oligodendrocyte formation and myelination in the same area (*Liu et al., 2012*; *Makinodan et al., 2012*; *Yamamuro et al., 2018*). We repeated this protocol, confirming the previously-described effect on circuit function by showing that isolated mice spent significantly less time than socially-experienced controls interacting with a novel mouse (*Figure 1A–C*). To quantify the myelination defects in these mice immediately following the isolation period we used a labelling strategy in which oligodendrocyte cell bodies, processes and myelin sheaths are revealed by CNPase immunostaining. The sparsely-myelinated layers II/III of the mPFC, a region myelinated during this isolation period (*Figure 1D*), enabled individual oligodendrocyte morphologies to be analysed for myelin sheath number and length (*Figure 1D–E*, *Figure 1—figure supplement 1*). Using this approach we showed that individual oligodendrocytes made fewer myelin sheaths (*Figure 1F–H*), although the length of the myelin sheaths formed by oligodendrocytes in these mice was unaffected (*Figure 1—figure supplement 2E–F*). Further, we found that the isolated mice also generated fewer oligodendrocytes in the mPFC (*Figure 1—figure supplement 2C–D*). In contrast, social isolation had no effect on myelin sheath number in the visual cortex (*Figure 1—figure supplement 3A–B*), demonstrating the region-dependent effects of this protocol on myelination.

Having validated our social isolation protocol by demonstrating experience-dependent changes in myelination, we next analysed EDN expression in the mPFC. We performed in situ hybridization studies examining the expression of all three EDN ligands (*Edn1-3*) in controls and following

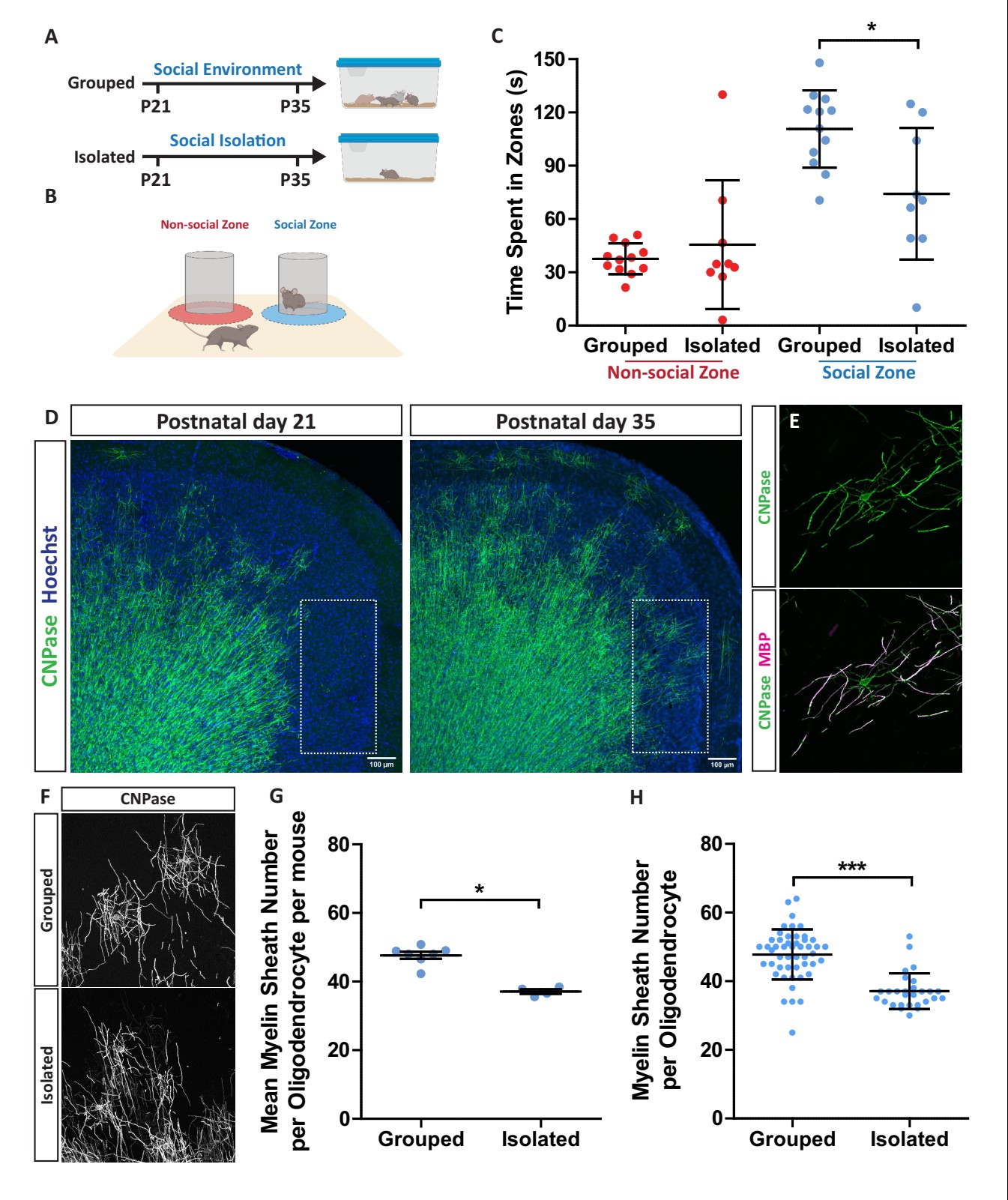

**Figure 1.** Social isolation in mice reduces layer II/III medial prefrontal cortex oligodendrocyte myelin sheath number. (**A**) Timeline for social isolation experiment. At postnatal day 21 male mice were housed in a social environment containing 3–5 mice or on their own in isolation. Mice were analysed at P35. (**B**) Schematic of social interaction assay. Mice were recorded for 5 min exploring an arena containing two identical wire mesh containers: one container housed an unrelated male wild type mouse (social zone), while the other remained empty (non-social zone). (**C**) Time spent within 2.5 cm of

*Figure 1 continued on next page*

*Figure 1 continued*

non-social container: Grouped 37.58 s ± 8.683 n = 12, Isolated 45.59 ± 36.28 n = 9 and social container: Grouped 110.7 s ± 21.71 n = 12, Isolated 74.24 s ± 37.07 n = 9 (mean ± standard deviation). Unpaired T-test p=0.0107. (D) Coronal section of mouse prefrontal cortex stained for CNPase and nuclei. Layers II/III of the medial prefrontal cortex outlined by dashed box. Scale bars = 100 µm. (E) Layer II/III oligodendrocyte stained for CNPase and MBP. (F) Representative images of medial prefrontal cortex oligodendrocytes stained for CNPase. (G) Mean number of myelin sheaths formed by oligodendrocytes per mouse. Grouped 47.66 ± 1.015 n = 7 mice, Isolated 37.11 ± 0.6425 n = 4 mice (mean ± standard error). Mann-Whitney test, p=0.0106. (H) Pooled data for number of myelin sheaths formed by layer II/III medial prefrontal cortex oligodendrocytes. Grouped 47.80 ± 7.289 n = 49 cells from seven mice, Isolated 37.11 ± 5.202 n = 28 cells from four mice (mean ± standard deviation). Mann-Whitney test, p=<0.001.

DOI: https://doi.org/10.7554/eLife.49493.002

The following figure supplements are available for figure 1:

**Figure supplement 1.** Z-stack through a CNPase positive oligodendrocyte.

DOI: https://doi.org/10.7554/eLife.49493.003

**Figure supplement 2.** Social isolation reduces oligodendrocyte generation in the mPFC.

DOI: https://doi.org/10.7554/eLife.49493.004

**Figure supplement 3.** Loss of oligodendroglial EDNRB reduces myelin sheath number in the visual cortex where social isolation does not affect myelination or *Edn1* expression.

DOI: https://doi.org/10.7554/eLife.49493.005

isolation. We confirmed robust *Edn1* and *Edn3* mRNA expression in controls, localised to laminin-positive blood vessels, which co-expressed the endothelial/pericyte marker *Pecam1* mRNA (*Figure 2A*) (*Yanagisawa et al., 1988*; *Hammond et al., 2014*). While previous studies have observed *Edn1* expression also localised to astrocytes following demyelination (*Hammond et al., 2014*) and microglia (*Zhang et al., 2014*), here we observed no *Edn1*, *Edn2* or *Edn3* mRNA in S100β positive astrocytes or IBA1 positive microglia (*Figure 2—figure supplement 1A–B*), indicating that endothelial cells and/or pericytes are the main source of *Edn* in the healthy mouse brain. Following social isolation, both the number of endothelial cells expressing *Edn1* and *Edn3* mRNA and the expression level of *Edn1* mRNA within each cell was significantly reduced in the mPFC (*Figure 2B–D*, *Figure 2—figure supplement 2F–G*). However, the vascular area and the number of cells expressing *Pecam1* mRNA were unaffected (*Figure 2—figure supplement 2A–E*). In contrast to the mPFC, there was no effect of social isolation on *Edn1* expression by *Pecam1* cells in the visual cortex (*Figure 1—figure supplement 3C*). We therefore conclude that the environmental deprivation associated with social isolation reduces vascular *Edn* production in the mouse mPFC.

## EDNRB loss reduces the number of myelin sheaths formed by individual oligodendrocytes in vivo

Next, we asked whether a reduction in EDN signalling could potentially contribute to the reduced levels of oligodendrocyte generation and myelination in the isolated mice. To do this, we defined the role of EDN signalling in myelination by performing genetic loss of function experiments to remove the relevant receptor in conditional knockout mice. We targeted the EDN receptor EDNRB due to the high levels of expression in forebrain oligodendrocytes and our own previous work identifying EDNRB as a regulator of myelination in vitro (*Yuen et al., 2013*; *Horiuchi et al., 2017*; *Marques et al., 2018*). We generated a conditional knock out (cKO) using a floxed *Ednrb* mouse line crossed with a line expressing cre recombinase driven by the *Pdgfra* promoter to delete *Ednrb* from oligodendrocytes and their precursor cells through development (*Figure 3—figure supplement 1A*). Comparison of the number of oligodendrocyte lineage cells in the mPFC in the control (*Pdgfra*-cre;*Ednrb*^wt/wt^) mice shown in *Figure 1* with EDNRB cKO (*Pdgfra*-cre;*Ednrb*^flox/flox^) mice revealed no significant difference in the oligodendrocyte progenitor cell (OPC) or oligodendrocyte generation (*Figure 3A*, *Figure 3—figure supplement 1B–D*). However, loss of EDNRB in the oligodendroglial lineage significantly reduced the number of myelin sheaths generated by individual oligodendrocytes (*Figure 3B–D*). EDNRB cKO oligodendrocytes demonstrated a 22% reduction in myelin sheath number compared to wild type littermates, but the lengths of the remaining sheaths were unchanged (*Figure 3—figure supplement 1E–F*). A similar effect of EDNRB loss on myelin sheath number was also seen in the visual cortex (*Figure 1—figure supplement 3A–B*). These experiments show that oligodendroglial EDNRB regulates the number of myelin sheaths formed by individual oligodendrocytes and, strikingly, that this effect of deleting EDNRB was identical to the

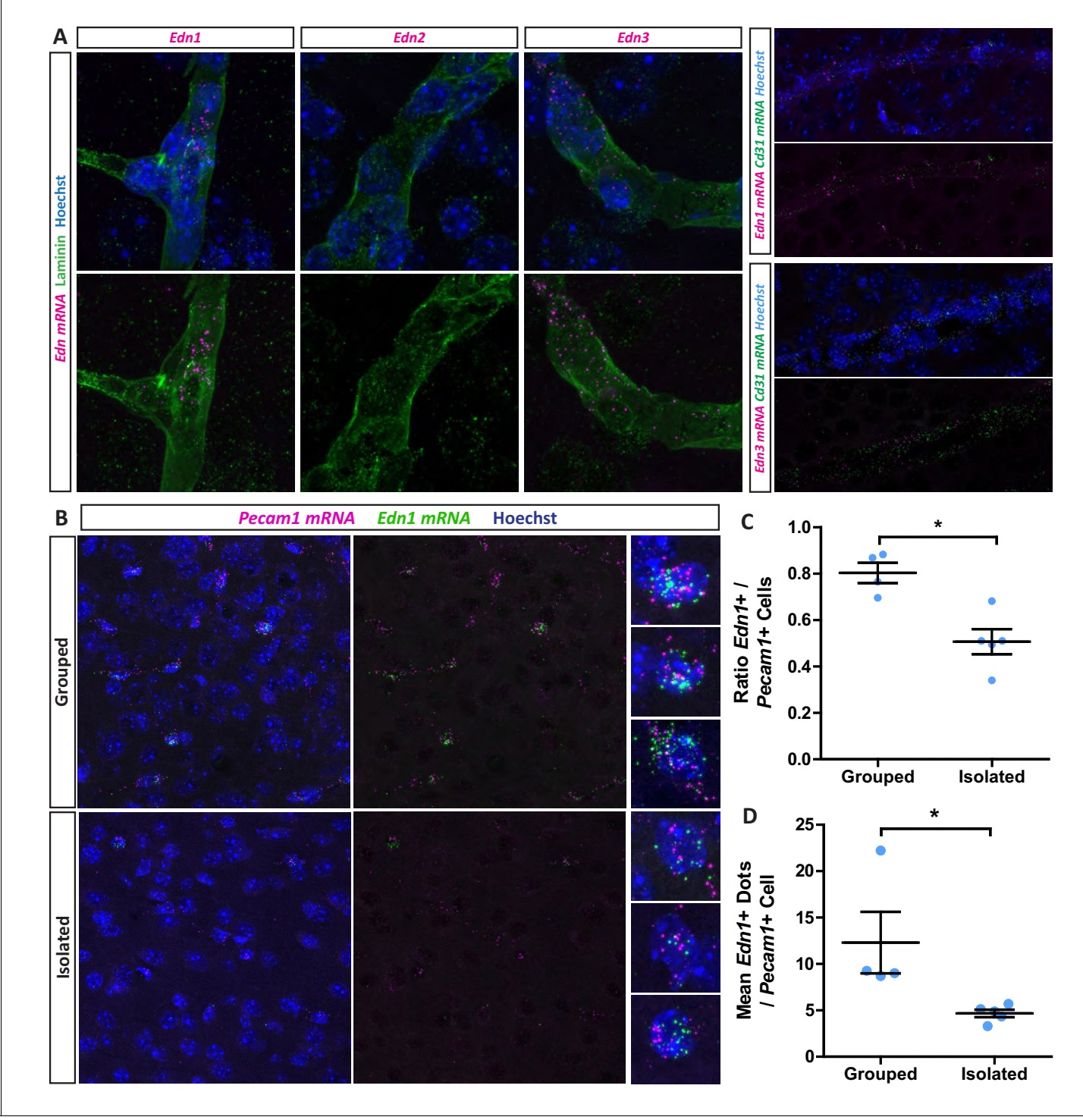

**Figure 2.** Social isolation reduces vascular endothelin expression. (**A**) *Edn1* and *Edn3* mRNA expression in laminin positive and *CD31* positive blood vessels as revealed by RNAScope in situ hybridisation. (**B**) Representative images of *Edn1* and *Pecam1* mRNA expression in the mPFC. (**C**) Quantification of the number of *Edn1* expressing *Pecam1* positive endothelial cells. Grouped 0.8033 ± 0.04411 n = 4 mice, Isolated 0.5074 ± 0.05412 n = 5 mice (mean ± standard error). Mann-Whitney test, p=0.0159. (**D**) Quantification of the mean *Edn1* mRNA molecules expressed by *Pecam1* positive cells per mouse. Grouped 12.29 ± 3.312 n = 4 mice, Isolated 4.673 ± 0.4059 n = 5 mice (mean ± standard error). Mann-Whitney test, p=0.0159.
DOI: https://doi.org/10.7554/eLife.49493.006

The following figure supplements are available for figure 2:

**Figure supplement 1.** EDN mRNA is not expressed in astrocytes and microglia.

*Figure 2 continued on next page*

*Figure 2 continued*

DOI: https://doi.org/10.7554/eLife.49493.007

**Figure supplement 2.** Social isolation does not affect medial prefrontal cortex vasculature.

DOI: https://doi.org/10.7554/eLife.49493.008

myelination deficits of individual oligodendrocytes within the mPFC following social isolation in wild-type animals.

By contrast, the numbers of oligodendrocytes within the mPFC differs between the two experimental manipulations – reduced by social isolation (*Figure 1—figure supplement 2C–D*) but not affected by deleting oligodendroglial EDNRB. Given these similarities and difference in the cellular phenotypes of social isolation and EDNRB deletion, we therefore next asked to what extent the changes in behaviour following isolation were phenocopied in the EDNRB cKO mice. Surprisingly, despite the additional effects of social isolation on oligodendrocyte number, the cell type specific loss of EDNRB from the oligodendrocyte lineage was sufficient to cause a significant reduction in the amount of time that cKO mice spent with the novel mouse in the social interaction task, although, as might have been expected, the effect size was smaller than in socially isolated mice, with a 33% reduction following social isolation and 20% following EDNRB cKO as compared to wild type mice (*Figure 3E*, *Figure 3—figure supplement 1G–H*). We conclude that loss of EDNRB signalling in oligodendrocyte lineage cells phenocopies a specific component of the response of oligodendrocytes to social isolation and that this, in turn, contributes to the behavioural deficits that result from social isolation.

## EDNRB signalling enhances myelin sheath number in 3D microfiber cultures

Given that cell-type specific loss of EDNRB results in oligodendrocytes making fewer myelin sheaths per cell, we wanted to ask whether activation of EDNRB signalling in oligodendrocytes might have the complementary effect and promote myelin sheath generation. To do this, we turned to a reductionist in vitro system in which myelination takes place in the absence of confounding influences from other cell types - a 3-dimensional microfiber culture system in which oligodendrocytes form myelin sheaths (*Lee et al., 2012a*; *Bechler et al., 2015*). In these cultures we tested both a peptide agonist (BQ3020) and a small molecule antagonist (BQ788) of the receptor. Rat oligodendrocyte precursors were seeded onto poly-l-lactic acid-coated microfibers and allowed to differentiate for 3 days before being treated with BQ3020 or BQ788. We saw a significant increase in the number of myelin sheaths produced by individual oligodendrocytes 11 days later in response to the EDNRB agonist (*Figure 4A–C*). Similar results were obtained with wild type mouse oligodendrocytes (*Pdgfra*-cre;*Ednrb*$^{wt/wt}$) while BQ3020 had no effect on EDNRB cKO (*Pdgfra*-cre;*Ednrb*$^{flox/flox}$) oligodendrocytes (*Figure 4F–G*), confirming the specificity of the agonist for EDNRB. In contrast to the agonist, treatment with the EDNRB antagonist had no effect on myelin sheath number in these cultures (*Figure 4A–C*), as might be expected given the lack of EDN-expressing blood vessel cells, and neither agonist nor antagonist altered sheath length (*Figure 4D*). We conclude from these experiments that the BQ3020 agonist selectively activates EDNRB-mediated signalling pathways in oligodendrocytes that increase myelin sheath number in vitro.

## EDNRB promotes myelin sheath formation through Protein Kinase C ε

The observation that we can stimulate myelination by oligodendrocytes in an EDNRB-dependent manner in vitro enabled a biochemical approach to identify the downstream signalling mechanisms by which this was mediated. We therefore next used a forward-phase phosphorylated-protein antibody array of mouse oligodendroglial cultures treated with the EDNRB agonist as an initial screen to identify these downstream pathways. As shown in *supplementary file 1*, one of the largest changes in protein phosphorylation following EDNRB activation was an increase in the phosphorylation of serine-729 of PKC epsilon, an isozyme of the Protein Kinase C (PKC) family. We confirmed an instructive role of PKCε in two ways. First, we examined myelination in oligodendrocyte microfiber cultures treated with FR 236924, a specific activator of PKCε, which promoted phosphorylation of PKCε at serine-729 (*Figure 5A*) and significantly increased myelin sheath formation (*Figure 5B–C*), similarly

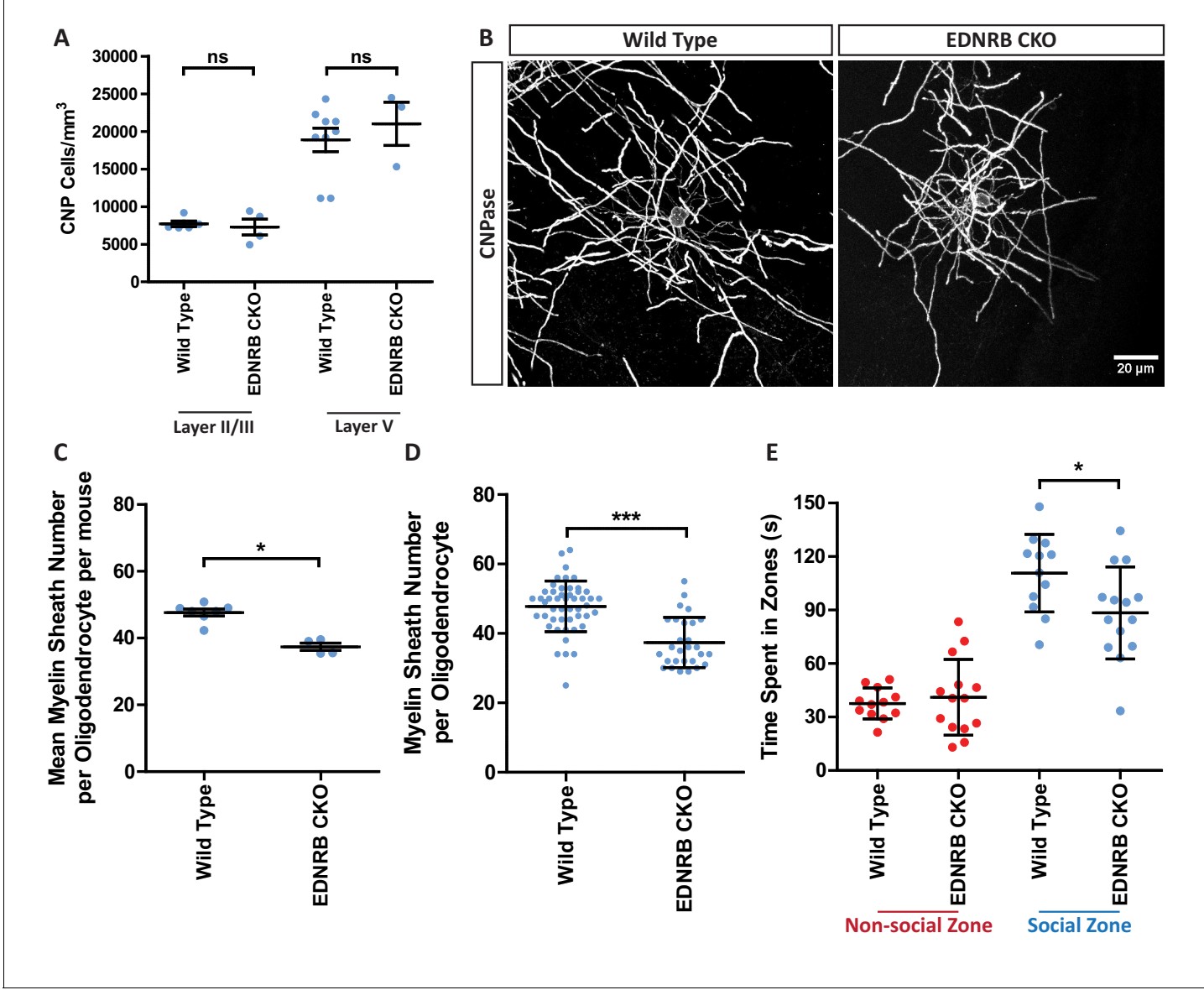

**Figure 3.** Loss of oligodendroglial EDNRB reduces myelin sheath number and reduces sociability. (**A**) Quantification of CNP positive cells in medial prefrontal cortex layers II/III: Wild type 7709 ± 378.7 n = 5 mice, EDNRB CKO 7288 ± 1054 n = 4 mice (mean ± standard error) and layer V: Wild type 18879 ± 1559 n = 9 mice, EDNRB CKO 21016 ± 2878 n = 3 mice (mean ± standard error). Mann Whitney test, layer II/III p=0.9048, layer V p=0.3527. (**B**) Representative images of medial prefrontal cortex oligodendrocytes stained for CNPase. Scale bar = 20 μm. (**C**) Mean number of myelin sheath formed by oligodendrocytes per mouse. Wild type 47.66 ± 1.015 n = 7 mice, EDNRB CKO 37.39 ± 1.099 n = 4 mice (mean ± standard error). Mann-Whitney test, p=0.0106. (**D**) Pooled data for number of myelin sheaths formed by layer II/III medial prefrontal cortex oligodendrocytes. Wild type 47.80 ± 7.289 n = 49 cells from seven mice, EDNRB CKO 37.39 ± 7.208 n = 28 cells from four mice (mean ± standard deviation). Mann-Whitney test, p=<0.001. (**E**) Time spent within 2.5 cm of non-social container: Wild type 37.58 s ± 8.683 n = 12, EDNRB CKO 41.06 ± 21.29 n = 14 and social container: Wild type 110.7 s ± 21.71 n = 12, EDNRB CKO 88.39 s ± 25.79 n = 14 (mean ± standard deviation). Unpaired T-test p=0.0267.

DOI: https://doi.org/10.7554/eLife.49493.009

The following figure supplement is available for figure 3:

**Figure supplement 1.** Conditional EDNRB knock out does not affect oligodendrocyte generation or myelin sheath length.

DOI: https://doi.org/10.7554/eLife.49493.010

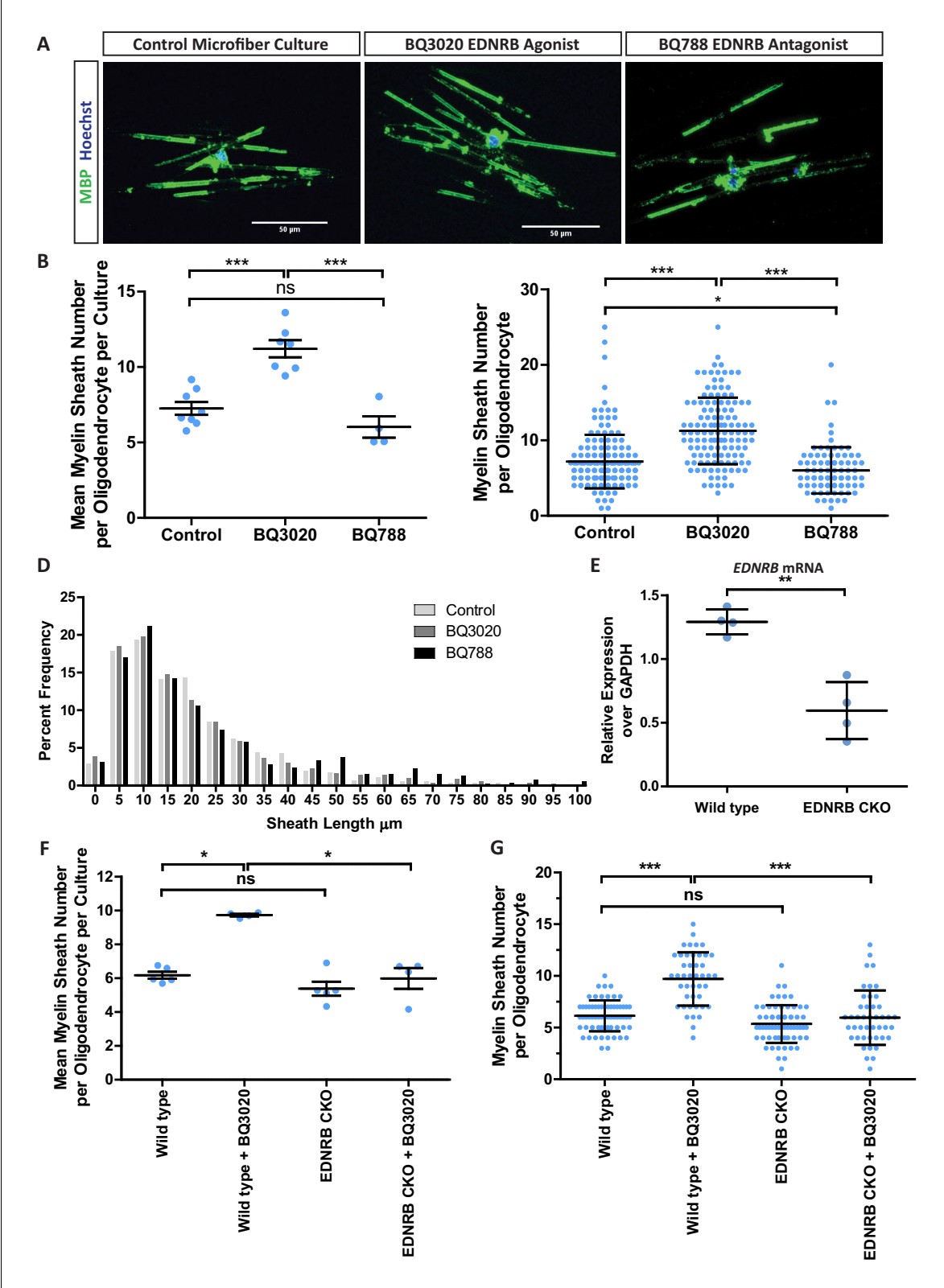

**Figure 4.** EDNRB enhances myelin sheath number in vitro. (**A**) Representative images of MBP positive oligodendrocytes in microfiber culture. Scale bar = 50 µm. (**B**) Mean number of myelin sheaths formed by rat oligodendrocytes on microfibers per independent culture preparation. Control 7.253 ± 0.4258 n = 8 independent cultures, BQ3020 11.21 ± 0.58635 n = 7 independent cultures, BQ788 6.026 ± 0.7046 (mean ± standard error) n = 4 independent cultures. 1-way ANOVA with Tukey's post hoc test. (**C**) Pooled data for number of myelin sheaths formed by rat oligodendrocytes on

*Figure 4 continued on next page*

Figure 4 continued

microfibers. Control 7.194 ± 3.544 n = 160 cells from eight independent cultures, BQ3020 11.25 ± 4.420 n = 127 from seven independent cultures, BQ788 6.024 ± 3.059 n = 85 cells from four independent cultures (mean ± standard deviation). Kruskal-Wallis test, with Dunns post hoc. (D) Frequency distribution of myelin sheath lengths formed on microfibers. (E) qPCR for EDNRB from mouse oligodendrocyte cultures. Wild type 1.292 ± 0.04933 n = 4 independent cultures, EDNRB CKO 0.5958 ± 0.1117 n = 4 independent cultures, BQ788 6.026 ± 0.7046 (mean ± standard error). Unpaired T-test p=0.0013. (F) Mean number of myelin sheaths formed by mouse oligodendrocytes on microfibers per independent culture preparation. Wild type 6.18 ± 0.2082 n = 5 independent cultures, Wild type + BQ3020 9.732 ± 0.07548 n = 4 independent cultures, EDNRB CKO 5.380 ± 0.4181 n = 5 independent cultures, EDNRB CKO + BQ3020 5.989 ± 0.6125 n = 4 independent cultures (mean ± standard error). 1-way ANOVA. (G) Pooled data for number of myelin sheaths formed by mouse oligodendrocytes on microfibers. Wild type 6.138 ± 1.499 n = 65 cells from five independent cultures, Wild type + BQ3020 9.705 ± 2.575 n = 44 from 4independent cultures, EDNRB CKO 5.345 ± 1.824 n = 65 cells from 5independent cultures, EDNRB CKO + BQ3020 5.955 ± 2.632 n = 44 cells from four independent cultures (mean ± standard deviation). Kruskal-Wallis test, with Dunns post hoc.

DOI: https://doi.org/10.7554/eLife.49493.011

to BQ3020. Second, we used zebrafish, due to their amenability for rapid pharmacological treatment and assessment, to test the prediction that the effects on myelination caused by the loss of EDNRB signalling would be rescued by activating the downstream target PKCε in vivo. We obtained a zebrafish line containing a mutation termed rose (rse) in one of the orthologues of *Ednrb (ednrba)*. Analysis of individual oligodendrocytes enabled by sparse labelling of these cells in the fish larvae revealed the same phenotype as in the EDNRB cKO mice; the average number of myelin sheaths formed per oligodendrocyte was reduced by 31% in *rse* homozygous fish (EDNRB Hom) compared to wild type controls (*Figure 5F–G*, *Figure 5—figure supplement 2A*), while sheath length was unaffected (*Figure 5—figure supplement 2B–C*). As predicted, activation of PKCε by FR 236924 treatment from 3 to 4 days post fertilisation, rescued myelin sheath number in *rse* homozygous fish (*Figure 5D–G*). This result confirms a role for PKCε downstream of EDNRB in the regulation of myelin sheath number in vivo.

## Increasing EDNRB signalling rescues the reduction in myelin sheath number resulting from social isolation

Our results have shown that social isolation decreases the levels of EDN expression in the CNS vasculature, and that EDNRB signalling in oligodendrocytes influences myelin sheath number. It follows that the decreased EDN signalling is likely contributing to the reduction in myelin sheath numbers seen in the medial prefrontal cortex following social isolation. If so, then we would predict that activation of EDNRB in vivo in the prefrontal cortex would rescue the myelin sheath phenotype associated with social isolation. To test this we used intranasal administration, a technique shown to enable the delivery of peptides into the CNS (*Scafidi et al., 2014*; *Crowe et al., 2018*), to introduce the EDNRB agonist BQ3020. Socially isolated mice were given 1 µg BQ3020 twice daily intranasal administration or saline for a period of 10 days from postnatal day 21 to 30 (during the period of isolation), and perfused at postnatal day 35 for analysis of mPFC myelination as above. Importantly, we confirmed that the daily handling involved in this protocol did not negate the effects of social isolation by showing that the sheath numbers in the control isolated mice administered saline had the same reduction in sheath numbers as completely isolated animals (*Figure 6B and D–E*). Strikingly, and in keeping with our prediction, 10 days of BQ3020 treatment to socially-isolated animals rescued the myelination phenotype normally seen in these mice, with a 20% increase in the number of myelin sheaths formed by mPFC oligodendrocytes in treated animals, meaning that the sheath numbers were now not significantly different to wild-type animals housed in groups (*Figure 6D–E*).

## Discussion

Here we identify a novel signalling pathway in which the oligodendroglial G-protein coupled receptor EDNRB regulates myelin sheath number and the response of oligodendrocytes to social experience (*Figure 6—figure supplement 2A*). We studied mice following a previously established experimental manipulation in which early postnatal social isolation leads to a reduction in the number of myelin sheaths formed by individual oligodendrocytes in the medial prefrontal cortex (mPFC) of mice. We showed that isolated animals have reduced EDN mRNA expression in the blood vessels of the mPFC. We found that perturbing endothelin signalling in oligodendroglia by a conditional

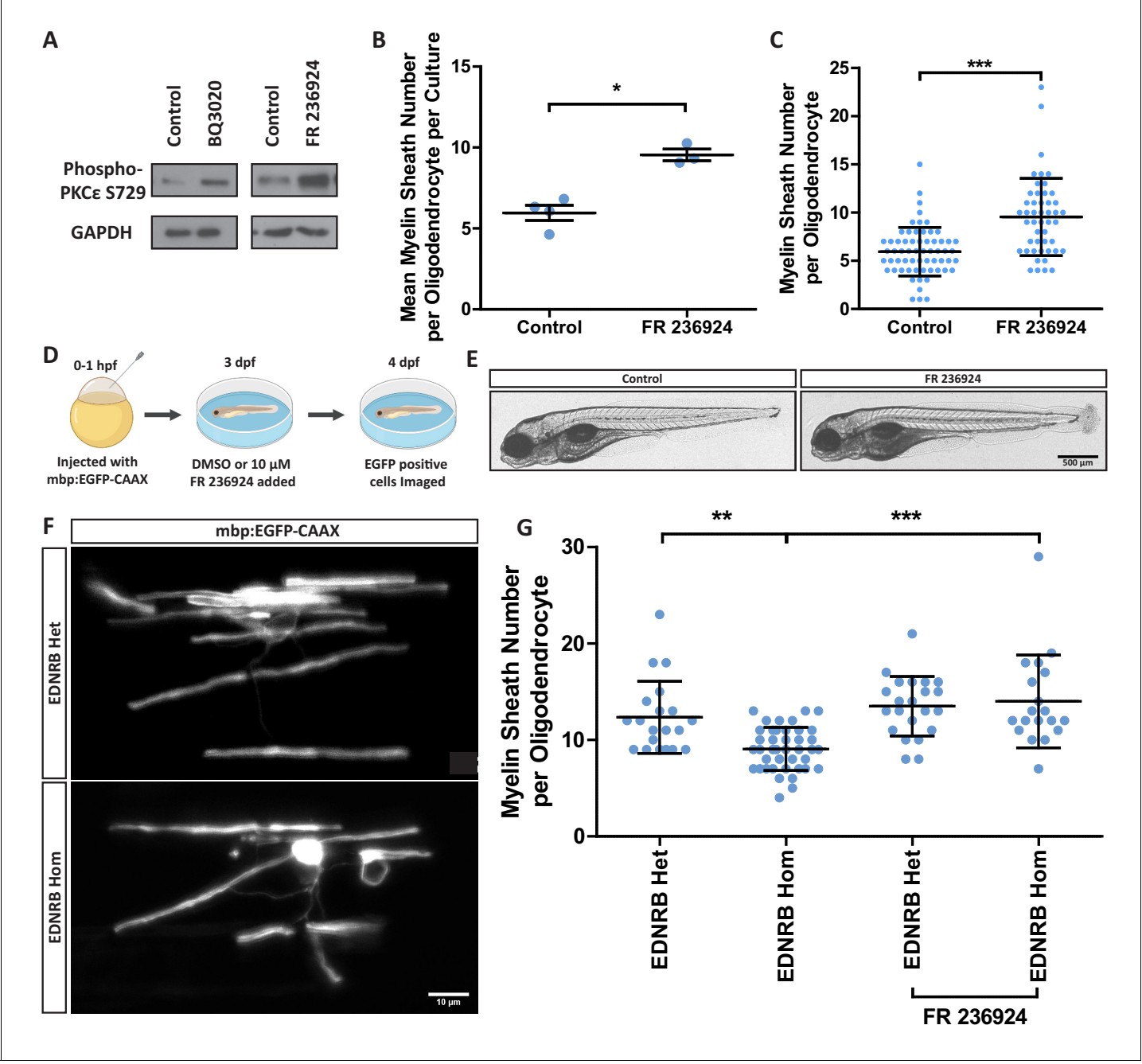

**Figure 5.** Protein kinase C epsilon is downstream of EDNRB to regulate myelin sheath number. (**A**) Western blot images of rat oligodendrocytes treated with EDNRB agonist BQ3020 and PKCε agonist FR 236924 for 15 min. Antibodies used: Phosphorylated PKCε S729 and loading control GAPDH. (**B**) Mean number of myelin sheaths formed by rat oligodendrocytes on microfibers per experiment. Control 5.959 ± 0.4708 n = 4, FR 236924 9.542 ± 0.3614 n = 3 (mean ± standard error). Unpaired T-test p=0.0024. (**C**) Pooled data for number of myelin sheaths formed by rat oligodendrocytes on microfibers. Control 5.952 ± 2.525 n = 62 cells from four experiments, FR 236924 9.542 ± 4.016 n = 48 from three experiments (mean ± standard deviation). Mann-Whitney test, p=<0.001. (**D**) Schematic for zebrafish larvae treatment with FR 236924. (**E**) Representative images of 4 dpf zebrafish larvae treated with DMSO control or FR 236924. Scale bar = 500 µm. (**F**) Representative images of mbp:EGFP-CAAX oligodendrocytes in four dpf zebrafish larvae. Scale bar = 10 µm. (**G**) Pooled data for number of myelin sheaths formed by zebrafish oligodendrocytes. EDNRB Het 12.35 ± 3.746 n = 20 cells, EDNRB Hom (*rse*) 9.073 ± 2.229 n = 41 from four experiments, EDNRB Het + FR 236924 13.5 ± 3.098 n = 22 cells from five experiments, EDNRB Hom + FR 236924 14 ± 4.807 n = 19 cells (mean ± standard deviation). 1-way ANOVA.

DOI: https://doi.org/10.7554/eLife.49493.012

The following figure supplements are available for figure 5:

**Figure supplement 1.** Global loss of EDNRB increases the number of oligodendrocytes in the zebrafish ventral spinal cord.

*Figure 5 continued on next page*

*Figure 5 continued*
DOI: https://doi.org/10.7554/eLife.49493.013
**Figure supplement 2.** Global loss of EDNRB reduces the number of myelin sheath formed by zebrafish oligodendrocytes.
DOI: https://doi.org/10.7554/eLife.49493.014
**Figure supplement 3.** Protein kinase C epsilon activation does not affect myelin sheath length.
DOI: https://doi.org/10.7554/eLife.49493.015

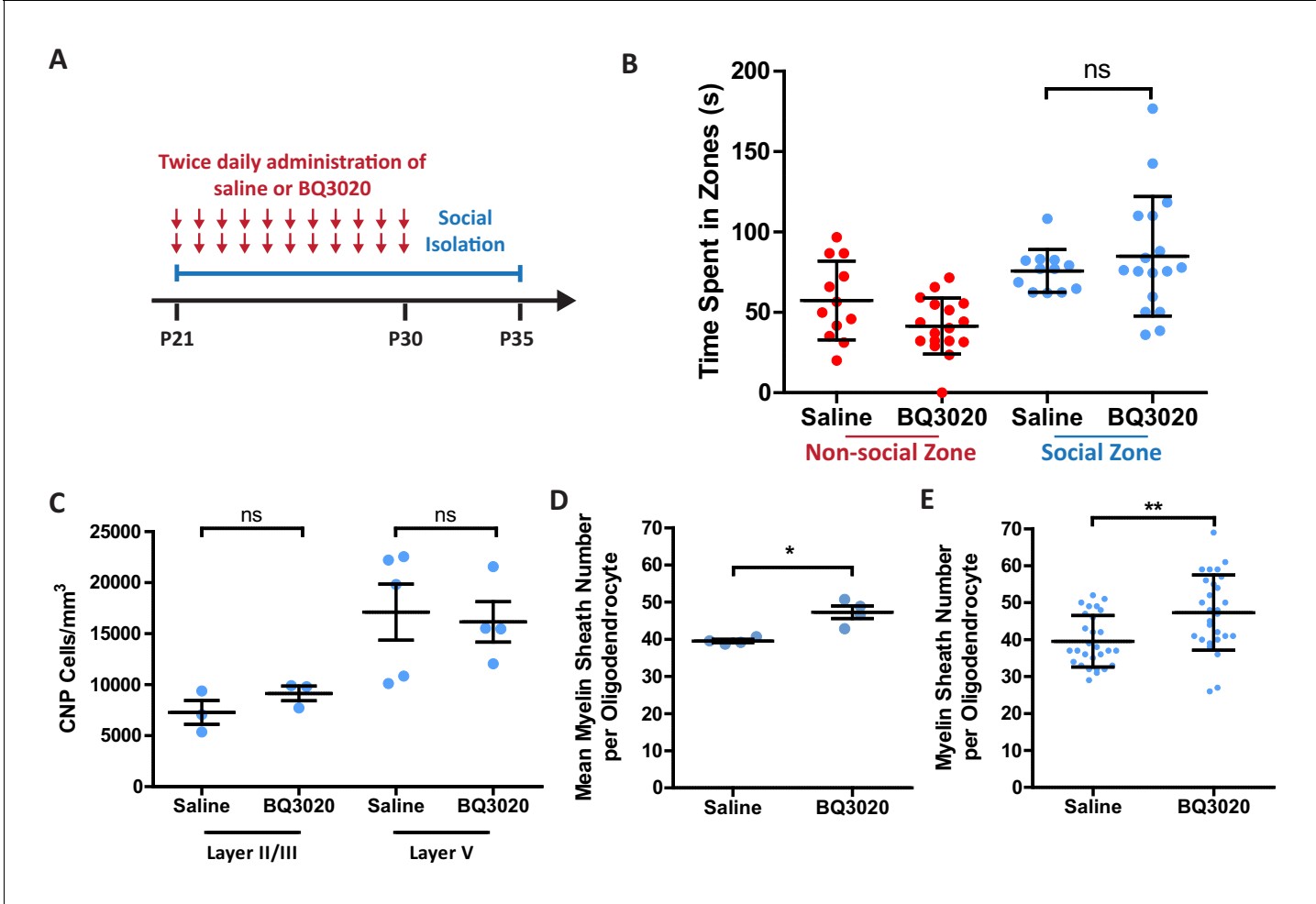

**Figure 6.** Intranasal administration of an EDNRB agonist rescues the myelin sheath number reduction caused by social isolation. (**A**) Timeline for the intranasal experiment. At postnatal day 21 male mice were housed on their own in isolation. Mice were given two daily administrations of saline or EDNRB agonist BQ3020 from P21-P30. Mice were analysed at P35. (**B**) Time spent within 2.5 cm of non-social container: Saline 57.42 s ± 24.44 n = 12, BQ3020 41.44 ± 17.43 n = 17 and social container: Saline 75.82 s ± 13.26 n = 12, BQ3020 84.96 s ± 37.20 n = 17 (mean ± standard deviation). T-test. (**C**) Quantification of CNP positive cells in medial prefrontal cortex layers II/III: Saline 7278 ± 1165 n = 3 mice, BQ3020 9142 ± 713.3 n = 3 mice (mean ± standard error) and layer V: Saline 17114 ± 2750 n = 5 mice, BQ3020 16154 ± 1980 n = 4 mice (mean ± standard error). (**D**) Mean number of myelin sheath formed by oligodendrocytes per mouse. Saline 39.54 ±0.4301 n = 4 mice, BQ3020 47.29 ± 1.687 n = 4 mice (mean ± standard error). Mann-Whitney test, p=0.0286. (**E**) Pooled data for number of myelin sheaths formed by layer II/III medial prefrontal cortex oligodendrocytes. Saline 39.54 ±6.973 n = 28 cells from four mice, BQ3020 47.29 ± 10.18 n = 28 cells from seven mice (mean ± standard deviation). Mann-Whitney test, p=0.0019.
DOI: https://doi.org/10.7554/eLife.49493.016

The following figure supplements are available for figure 6:

**Figure supplement 1.** Intranasal administration of EDNRB agonist BQ3020 does not affect myelin sheath length.
DOI: https://doi.org/10.7554/eLife.49493.017
**Figure supplement 2.** Proposed model for how EDNRB regulates myelin sheath number.
DOI: https://doi.org/10.7554/eLife.49493.018

knockout of the EDN receptor EDNRB phenocopies the reduced number of sheaths seen in the isolated animals. In turn intranasal administration of an EDNRB agonist during the period of isolation rescues the effects of social deprivation on myelination. Together with additional experiments in cell culture and zebrafish, which identified the protein kinase C (PKC) epsilon isoform as being activated downstream of EDNRB to regulate myelination by oligodendrocytes, our loss- and gain-of-function experiments demonstrate a role for endothelin signalling in regulating the number of myelin sheaths formed by individual oligodendrocytes.

Our work provides evidence for the importance of the interaction between the vasculature and oligodendroglia during myelination per se, extending observations indicating that the vasculature can influence earlier stages of the oligodendrocyte lineage. For example, during development, oligodendrocyte progenitor cells use blood vessels as a scaffold to migrate along (*Tsai et al., 2016*). Once in place, local oxygen levels influence their differentiation into oligodendrocytes, stalling the process when levels are low. Only when an adequate oxygen supply is established are the cells able to differentiate and begin the process of myelination (*Yuen et al., 2014*). Here, we demonstrate a third role – the regulation of myelination itself through the control by EDN of myelin sheath number by individual differentiated oligodendrocytes. Such a role is potentially important, as it provides an indirect mechanism by which the regulation of myelination in the CNS might be linked to levels of neuronal activity. A key function of myelin is the provision of metabolic substrates to the underlying axon via specific transporters within the inner layer of the sheath (*Fünfschilling et al., 2012*; *Lee et al., 2012b*; *Meyer et al., 2018*). These substrates drive ATP production within the axon, so providing a source of energy at some distance from the cell body as required to sustain axonal activity. Prior work has implicated that NMDA receptor activation in oligodendrocytes following glutamate release by active axons in relaying the need for metabolic support of axons to oligodendrocytes (*Saab et al., 2016*). We now propose that an increase in EDN production by endothelial cells, as predicted to occur in the presence of increased blood flow or hypoxia, will increase myelin sheath generation by local oligodendrocytes and so provide a further means of linking energy supply and demand. Changes in blood flow and hypoxaemia are associated with increased cortical activity (forming the basis of the BOLD signal detected by fMRI). Therefore, an EDN-mediated pathway driving an increase in myelination is a plausible mechanism for ensuring that the metabolic demands of active axons are met. Further work examining the effect of EDNRB loss in oligodendrocytes on axonal function and viability in ageing are required to test this further.

A second important implication of our study relates to the emerging concept of adaptive myelination. This concept argues that myelination in the CNS is plastic, and regulated by differing codes of activity, each eliciting different effects on myelination that in turn alter circuit function. This hypothesis therefore proposes that adaptive myelination represents a form of neural plasticity that could, like synaptic plasticity, enable the brain to modify circuit function in response to experience-in other words, to adapt. However, this hypothesis remains to be fully explored, and a key requirement for these experiments is the identification of the mechanisms that link changes in activity patterns following experience to the changes in oligodendrocyte number and/or sheath formation that could alter circuit function. Prior work has identified a direct role for axon-derived signals in regulating myelination. We now identify a mechanism by which the effects of activity could lead to an increased number of myelin sheaths through indirect signalling via the vasculature. Our findings showing that social isolation reduces endothelial EDN expression, that an oligodendroglial-specific deletion of the relevant EDNRB receptor phenocopies the myelination defect caused by social isolation, and that an EDNRB agonist rescues the myelination defect of individual oligodendrocytes in the mPFC together argue strongly for a role of vascular EDN synthesis in mediating such indirect effects. Our results do not however imply that changes in EDN signalling are solely responsible for the effects of social isolation on myelination. It is clear, for example, that social isolation can affect oligodendrocyte number, a feature that does not appear to be controlled by EDN in the healthy nervous system. Other pathways could contribute to this aspect of adaptive myelination. Such potential pathways are BDNF signalling via oligodendroglia TrkB (*Geraghty et al., 2019*; *Gibson et al., 2019*) and glutaminergic signalling via AMPAR-mediated effects on newly differentiating oligodendrocytes (*Kougioumtzidou et al., 2017*). Further work manipulating EDN expression specifically in endothelial cells during social isolation is required to explore this.

Based on recent live imaging studies showing that myelin sheath numbers in cortical oligodendrocytes remain largely stable throughout life (*Hill et al., 2018*; *Hughes et al., 2018*), we propose that

any endothelin-mediated regulation of sheath number is likely to occur around the birth of each newly formed oligodendrocyte. The continuous generation of oligodendrocytes in the adult CNS, itself increased by activity-related signals ensures that dynamic regulation of endothelin signalling from the vasculature has the capacity to play a role in adaptive myelination throughout life.

How though might new myelin sheaths formed by individual oligodendrocytes in response to EDN signalling enhance circuit function the cortex? By linking metabolic demand to metabolic support for axons in regions of high activity (as discussed above), EDN signalling in cortical oligodendrocytes could play a central role in enhancing the ability of the axon to sustain higher levels of energy-requiring conduction, so enabling changes in circuit function. An intriguing possibility suggested by a comparison of gene expression in cortical oligodendroglia and spinal cord oligodendroglia showing higher levels of expression of EDNRB in the former (*Horiuchi et al., 2017*; *Marques et al., 2018*), is that these cortical oligodendrocytes are specialised for this role linking metabolic demand to support. By contrast, oligodendrocytes in white matter might be more specialised for rapid axonal conduction. Further work examining oligodendrocyte heterogeneity between grey matter and white matter regions of the CNS will be required to test this. Another way in which additional myelin sheaths might impact circuit function is the addition of sheaths to axons in the cortex that are discontinuously and sparsely myelinated (*Tomassy et al., 2014*; *Hill et al., 2018*; *Hughes et al., 2018*). The resulting effects on conduction velocity resulting from short stretches of axons now supporting rapid saltatory conduction could, as has been suggested elsewhere, enable activity-dependent myelination to have a role in signal synchrony (*Seidl et al., 2010*; *Seidl et al., 2014*; *Freeman et al., 2015*; *Baraban et al., 2016*; *Timmler and Simons, 2019*). However, such a model requires a very precise link between axonal selection and myelination, and the precision required may be difficult to achieve with a diffusible vascular-derived signal.

Our findings on the role of EDN in myelination complement other investigations demonstrating a role of astrocytic EDN hindering oligodendrocyte differentiation during remyelination (*Hammond et al., 2014*; *Hammond et al., 2015*). In agreement with this observation, our global EDNRB mutant zebrafish were found to have an increased number of oligodendrocytes (*Figure 5—figure supplement 1*). Together previous studies and ours suggest that EDN signalling plays two roles, indirectly inhibiting oligodendrocyte formation, but, promoting myelin sheath formation once differentiation has occurred. At apparent odds with this conclusion, conditional loss of EDNRB from oligodendrocyte progenitors had no effect on remyelination; having deleted OPC EDNRB from adult mice 3 days prior to demyelination of the external capsule, Hammond and colleagues observed that the percentage of remyelinated axons, assessed by electron microscopy, was unchanged 14 days after injury (*Hammond et al., 2015*). However, this divergence may result from the different microenvironments of developmental myelination and remyelination or from intrinsic differences between cortical and white matter oligodendrocytes, as discussed above, and further studies examining grey matter remyelination are required to resolve the point.

The conclusion that EDN has contrasting effects on oligodendrocyte formation and on myelination highlights that these two steps in oligodendrocyte development are regulated independently. This in turn has important implications for the development of regenerative therapies for the progressive neurodegenerative phase of the demyelinating disease Multiple Sclerosis (MS). Current strategies seeking to re-purpose FDA approved drugs to enhance remyelination, and so restore the neuroprotective effect of the myelin sheath to the axon, have focused largely on the differentiation step from precursor cell to oligodendrocyte, with one screen using micropillar arrays to examine the later stage of wrapping (*Mei et al., 2014*). None however specifically examine the key final step-formation of the sheath itself. Neuropathological studies of MS lesions revealing pre-myelinating oligodendrocytes unable to complete remyelination suggest for that failure of this final step may contribute to pathology. These findings emphasise that the already differentiated oligodendrocyte represents a possible target for remyelination strategies in MS, a conclusion reinforced by recent papers using Carbon14 dating, electron microscopy or snRNA seq of post mortem human MS material providing evidence that pre-existing adult oligodendrocytes contribute to repair (*Duncan et al., 2018*; *Jäkel et al., 2019*; *Yeung et al., 2019*). Strategies to identify targets within pathways such as that activated by EDNRB that promote sheath formation directly will therefore be an important addition to the current approaches being taken in drug discovery for progressive multiple sclerosis.

# Materials and methods

## Key resources table

| Reagent type (species) or resource | Designation | Source or reference | Identifiers | Additional information |
|---|---|---|---|---|
| Genetic reagent (*M. musculus*) | Ednrb flox/flox | The university of Edinburgh, *Bagnall et al., 2006*; *Ge et al., 2006* | | |
| Genetic reagent (*M. musculus*) | Pdgfra-cre | Jackson labs | #013148 | |
| Genetic reagent (*D. rerio*) | Rse Tlf802 | *Frohnhöfer et al., 2013*; *Krauss et al., 2014* | | |
| Genetic reagent (*D. rerio*) | Tg(mbp: EGFP) | The University of Edinburgh, *Almeida et al., 2011* | | |
| Antibody | Mouse monoclonal anti CNPase | Atlas | AMAb91072 | 1:2000 |
| Antibody | Rat monoclonal anti MBP | Serotec | MCA409S | 1:250 |
| Antibody | Rat monoclonal anti PECAM1 | BD Pharmingen | 550274 | 1:100 |
| Antibody | Rabbit monoclonal anti S100β | Thermo | MA5-12969 | 1:100 |
| Antibody | Rabbit monoclonal anti IBA1 | Abcam | ab178846 | 1:500 |
| Antibody | Rabbit polyclonal anti Laminin | Abcam | ab11575 | 1:300 |
| Antibody | Rabbit polyclonal anti OLIG2 | Millipore | ab9610 | 1:100 |
| Antibody | Mouse monoclonal anti CC1 | Abcam | ab16794 | 1:300 |
| Antibody | Mouse monoclonal anti GAPDH | Millipore | MAB374 | 1:1000 |
| Antibody | Mouse monoclonal anti Beta actin | Abcam | ab 8226 | 1:1000 |
| Antibody | Rabbit polyclonal anti, Phospho-PKCε S729 | Abcam | 88241 | 1:1000 |
| Commercial assay, kit | Phospho-explorer antibody array | Full Moon Biosystems, | Phospho-explorer array (PEX100) | |
| Sequence-based reagent | *Edn1* | Advanced Cell Diagnostics | 435221 | |
| Sequence-based reagent | *Edn2* | Advanced Cell Diagnostics | 418221 | |
| Sequence-based reagent | *Edn3* | Advanced Cell Diagnostics | Custom made | |

*Continued on next page*

*Continued*

| Reagent type (species) or resource | Designation | Source or reference | Identifiers | Additional information |
|---|---|---|---|---|
| Sequence-based reagent | *Pecam1* | Advanced Cell Diagnostics | 316721-C3 | |
| Sequence-based reagent | *Olig2* | Advanced Cell Diagnostics | 447091 | |
| Sequence-based reagent | *Ednrb* | Advanced Cell Diagnostics | 473801 | |
| Peptide, recombinant protein | BQ3020 | Tocris | 1189 | 100 ng/mL |
| Chemical compound, drug | BQ788 | Tocris | 1500 | 100 ng/mL |
| Chemical compound, drug | FR236924 | Tocris | 0373 | 25 µM |
| Other | Microfibers | The Electrospinning company | | 1–2 mcro diameter poly-l-lactic acid |
| Software, algorithm | Any-maze software | http://www.anymaze.co.uk/ | | |
| Software, algorithm | ImageJ | https://imagej.nih.gov/ij/ | | |
| Software, algorithm | Graphpad Prism | https://www.graphpad.com/scientific-software/prism/ | | |

## Mice

Animal husbandry and experiments were performed under UK Home Office project licenses issued under the Animals (Scientific Procedures) Act. *Ednrb*<sup>flox/flox</sup> (**Bagnall et al., 2006**; **Ge et al., 2006**) were generously provided by Professor David Webb and Professor Yuri Kotelevtsev (Edinburgh University) where exons 3 and 4 are flanked with Cre-LoxP sites. Homozygous mice for the EDNRB floxed allele were crossed to *Pdgfra*-cre mice obtained from Jackson laboratories (013148). Offspring were then backcrossed to create mice Heterozygous for the floxed allele and carriers for the Cre transgene. Experimental mice were obtained by crossing animals heterozygous for the floxed *Ednrb* allele and carriers of the Cre transgene generating both *Ednrb* wild type control and *Ednrb* floxed homozygous mice in the same litters. Mice were genotyped by transnetyx and confirmed as a CKO by performing RNAScope for *Ednrb* (described below) Mice of each genotype were used at the ages and in the numbers stated in the Results. For the social isolation experiments, postnatal day 21 male mice were either group housed in a regular social environment, containing 3–5 mice, or in isolation, for 2 weeks.

## Sociability test

At postnatal day 35 mice were allowed to freely explore an open field arena for 5 min containing two identical wire mesh containers. One container housed an unrelated male wild type mouse of a similar age, while the other remained empty. Mice were allowed to explore the arena for 5 min and the duration of time during which the mouse of interest came within 2.5 cm of either container was automatically recorded using Any-maze software. Speed and distance travelled were also recorded by the software. In all tests, mice were assessed with the experimenter blind to genotype.

## Intranasal administration

Mice were given brief isoflurane inhalation anaesthesia before being held upside down. Using a pipette, 5 µL of either saline or 1 µg BQ3020 (Tocris - 1189) in saline was placed on either nostril. This was repeated twice daily (9am and 5pm) from postnatal day 21 to postnatal day 31.

## Antibodies

CNPase Atlas AMAb91072 (1:2000), MBP Serotec MCA409S (1:250), PECAM1 BD Pharmingen 550274 (1:100), S100ß Thermo MA5-12969 (1:100), IBA1 Abcam ab178846 (1:500), Laminin Abcam ab11575 (1:300), Olig2 Millipore AB9610 (1:100), CC1 Abcam ab16794 (1:300).

## Immunofluorescence staining - Cryosections

Animals were intracardially perfused with 4% PFA (wt/vol; Sigma) in PBS, after which brains were post-fixed overnight and then cryoprotected in sucrose prior to embedding in OCT and storage at −80˚C. Brains were cryosectioned coronally at a thickness of 16 µm using a Thermo cryostat and mounted onto Superfrost Plus slides. Sections were blocked for 1 hr at room temperature in 10% goat serum, 0.1% triton in PBS. Primary antibodies were incubated in block solution at 4˚C overnight. Sections were washed with PBS for 3 × 15 min at room temperature and stained using species-specific Alexa fluorophore-conjugated antibodies in block solution for 1 hr at room temperature. Sections were washed in PBS for a further 3 × 15 min and stained with Hoechst for 5 min and mounted onto slides with fluoromount G. Mouse medial prefrontal cortex was defined as the infralimbic and prelimbic areas between bregma 1.7 mm and 2 mm and visual cortex bregma −3.5 mm and −4.5 mm.

## Immunofluorescence staining - Vibratome Sections for Oligodendrocyte morphological analysis

Animals were intracardially perfused with 4% PFA (wt/vol; Sigma) in PBS, after which brains were post-fixed overnight and embedded in 2% low melting point agarose. Using a Leica vibratome, 100 µM coronal free-floating sections were cut. Sections underwent antigen retrieval in 0.05% Tween20, 10 mM tri-sodium citrate (pH 6.0) at 95˚C for 20 min. Sections were then blocked for 3 hr at room temperature in 10% goat serum, 0.25% triton in PBS. Primary antibodies were incubated in block solution at 4˚C on a rocker for 24 hr. Sections were washed with PBS for 3 hr at room temperature and stained using species-specific Alexa fluorophore-conjugated antibodies in block solution for 4 hr at room temperature. Sections were washed in PBS for a further 3 hr and stained with Hoechst for 20 min and mounted onto slides with fluoromount G. Sections were analysed with the experimenter blind to experimental condition and/or genotype as below.

To analyse oligodendrocyte number 10 fields of 40x magnification were taken from 2 100 µm sections per mouse of cortical layer II/III or layer V using a SP8 confocal microscope. CNP positive, Olig2 and CC1 cells were counted.

To analyse oligodendrocyte morphology, random areas of the medial prefrontal cortex layer II/III were imaged at 63x magnification using an SP8 confocal microscope. seven individual CNP positive oligodendrocytes per mouse, each with all myelin sheaths present within the 100 µm section as assessed by following each process from the cell body and ensuring none exited the section, were imaged using a z step size of 0.5 µm. Analysis was performed using an ImageJ plugin – simple neurite tracer (*Longair et al., 2011*).

## RNAScope In situ hybridisation

Cryosections (cut as above) were processed as recommended by Advanced Cell Diagnostics. Briefly sections were dried overnight at 60˚C, after which they were incubated at 40˚C with pre-treatment 4 for 30 min before incubation with RNAScope probes for 2 hr at 40˚C. RNAScope probes used were; *Edn1* (435221), *Edn2* (418221), *Edn3* (custom made), *Pecam1* (316721-C3), *Oolig2* (447091-C2), *Ednrb* (473801). Following the RNAScope protocol, sections were stained as above.

For analysis, 5 random 63x areas of medial prefrontal cortex layer II/III were imaged using a SP8 confocal microscope. Cells positive for *Pecam1* and *Edn1/3* were counted. Sections were analysed with the experimenter blind to experimental condition and/or genotype.

## Rat oligodendrocyte precursor cell culture

Rat OPCs were prepared from mixed glial cultures as described previously (*McCarthy, 1980*; *Bechler et al., 2015*). Briefly, cortices of postnatal day 0–2 Sprague Dawley rats were dissected out. The tissue was digested with 1.2 Units/mL papain, 0.1 mg/mL L-cysteine and 0.40 mg/mL DNase for 1 hr at 37°C. Tissue was cultured in DMEM, 10% FCS, 1% P/S in T75 flasks, pre-coated with 5 µg/mL poly-D-lysine, at a density of 1.5 brains per flask. Cells were grown at 37°C in 7.5% $CO_2$ with medium changes every 2–3 days. After 10–12 days cells were mechanically separated on an orbital shaker at 250 rpm, 37°C. Loosely attached microglia were removed by shaking for 1 hr. Further shaking for 16–18 hr detached OPCs. Cell yield was counted using a haemocytometer and plated in assay-dependent conditions as below.

## Microfiber cultures

Custom parallel-aligned microfibers were purchased from the Electrospinning Company. 1–2 micron diameter poly-l-lactic acid microfibers were synthesised and suspended over plastic scaffolds fitting into 12-well tissue culture plates. Microfibers were washed with 70% EtOH for 10 min followed by coating with PDL for 1 hr at 37°C in a 12 well tissue culture plate. Microfibers were washed twice with sterile water and left in preheated myelination media. 35,000 rat OPCs and 50,000 mouse OPCs in myelination media (50:50 DMEM:Neurobasal Media, B27 (Invitrogen), 5 µg/mL N-acetyl cysteine, and 10 ng/mL D-biotin, ITS, and modified Sato (100 µg/mL BSA fraction V, 60 ng/ml Progesterone, 16 µg/ml Putrecsine, 400 ng/mL Tri-iodothyroxine, 400 ng/mL L-Thyroxine; reagents from Sigma-Aldrich)) were triturated to break up cell clumps and added dropwise to the microfibers. Cells were left to recover for 3 days before media changing and addition of treatment, followed by subsequent media changes every 3 days.

After 14 days of culture cells were fixed in 4% PFA for 15 min. To visualise myelination, cells were permeabilised with 0.1% Triton-X for 10 min and stained with for MBP (1:250) overnight at 4°C followed by incubation with Alexa 488-conjugated goat ant-rat (1:1000) for 1 hr at room temperature and Hoescht for 5 min to visualise nuclei.

To analyse myelination 15–30 individual myelinating oligodendrocytes from one coverslip were imaged (with the experimenter blind to condition) using an SP8 confocal at 40x magnification with a z-step of 0.35 µm. The same settings (laser power, gain, offset etc.) were used between coverslips. ImageJ was used to analyse myelination, again with the experimenter blind to condition. A sheath was defined as a continuous MBP positive wrap fully surrounding a microfiber as assessed using the 0.35 µm z-series. Concentric tubes were traced and the length measured. In addition the number of concentric sheaths made per individual oligodendrocyte was recorded. Sheath lengths were grouped into 5 µm bins and the frequency from one experiment calculated. Mean frequencies from at least three experiments were generated and plotted as a frequency distribution.

Drugs used: BQ3020 (100 ng/mL, Tocris - 1189) in 0.03M sodium bicarbonate, BQ788 (100 ng/mL, Tocris - 1500) in DMSO, FR236924 (25 µM, Tocris - 0373) in DMSO.

## Mouse oligodendrocyte precursor cell culture

Mouse OPCs were isolated from P6-P9 pups as described (*Watkins et al., 2008*). Ear clips were taken for subsequent genotyping. Briefly, cerebral cortices were dissected, diced and dissociated into single-cell suspensions gently using MACS Neural Tissue Dissociation Kit P (130-092-628). Cells were resuspended in 0.2% BSA, Insulin, PBS and transferred to treated tissue culture dishes coated with BSL1 (L-1100, Vector Labs) for 15 min twice. Cell solutions were then transferred to dishes coated with anti-PDGFRα (CD140a) for 45 min. Solutions were aspirated and attached cells washed twice with media and removed with a cell scraper. All collected cells were added to vented T75 flasks and grown at 37°C 7.5 $CO_2$. Cells were grown in myelination media containing PDGF and NT3 and changed every 2 days and supplemented daily with PDGF. After 7–9 days confluent flasks were washed with PBS and then detached using TrypLE for 10 min at 37°C. Solutions were centrifuged at 1000 rpm for 5 min, resuspended and counted using a haemocytometer.

## qPCR

75,000 mouse OPCs were cultured on pre-coated PDL six well plates for 2 days. RNA was extracted from cells using a Qiagen RNeasy mini kit. RNA was then converted to cDNA libraries using a

SuperScript First-Strand Synthesis System. Sybr green qPCR was performed using primers either bought from Qiagen or designed at 0.5 µM. qPCR was performed on a LightCycler 480 II. CT values from designed primers were normalised against GAPDH purchased from Qiagen.

## Full moon Phospho-Explorer antibody array

Processing of the phospho-explorer array (PEX100) was performed following the Full Moon Biosystems guidelines. OPCs were immunopanned from wildtype mice and expanded in the presence of PDGF. Upon confluency, OPCs were differentiated through supplementation of CNTF, NT3 and T3. After 2 days of differentiation cells were starved for 4–5 hr in media devoid of supplementation to remove stimulation from the media and then treated for 15 min with either vehicle or BQ3020 (100 ng/mL). Cells were lysed in RIPA buffer containing protease and phosphatase inhibitors (Calbiochem – 539134 and 524621) and processed for the array as recommended by Full Moon Biosystems.

Briefly, proteins present in the lysates were biotinylated through incubation with biotin. Chips were blocked using milk and biotinylated protein lysates were washed over. Binding of the individual proteins to each antibody spot on the array was assessed through fluorescent labelling with a dye-labelled streptavidin read using an Innopsys 710-IR scanner and analysed using Mapix software.

For each antibody the background intensity was subtracted, dye signal normalised and an average calculated of the duplicate spots. The ratio was calculated of binding to the phosphorylated amino acids vs the binding to the non-modified regions of the protein for each molecule, calculating this for both control and BQ3020 treated cells. The fold change in phosphorylation for each targeted amino acid was generated by comparing BQ3020 to vehicle. For selection a fold change of greater than two and less than 0.5 was set as the cut-off.

Antibody array was performed once – one cell lysate per condition.

## Western blot

For western blot analysis, 9 cm treated plastic tissue culture dishes were coated with PDL for either 1 hr at 37°C or overnight at room temperature. 1 million OPCs were added in myelination media to coated plates. After 3 days in culture, oligodendrocytes were starved from media supplementation through incubation with DMEM and 1% P/S only for 4–5 hr. Fresh starvation media was then added containing either vehicle control, BQ3020 (100 ng/mL) or FR236924 (25 µM). After 10 minutes cells were lysed and scraped into RIPA buffer with protease and phosphatase inhibitors (Calbiochem – 539134 and 524621) for 10 min on ice. Lysates were spun at 16,000 g for 10 min and supernatants retained.

Protein concentration was estimated using a BCA assay kit and loaded into precast protein gels with a protein marker of known molecular weights. Gels were ran at 60 volts for 30 min and then increased to 100 volts for 1 hr. Protein was transferred from the gels to nitrocellulose membranes pre-treated with methanol at 400 mA for 2 hr on ice. Membranes were blocked in 4% BSA in TBS-0.1% Tween for 40 min and incubated with primary antibodies (GAPDH Millipore MAB374, Beta-actin Abcam ab8226, Phospho-PKCε S729 Abcam 88241) overnight at 4°C on an orbital shaker. Membranes were washed in TBS-Tween for 30 min and incubated with species specific secondary horseradish peroxide antibodies at room temperature for 1 hr. Membranes were washed for a further 30 min and incubated with ECL2 for 5 min. Blots were detected using a Licor scanner. Subsequent western blots were performed using the same membranes following removal of bound antibodies through incubation with stripping buffer for 15 min.

## Zebrafish

Animal husbandry and experiments were maintained in accordance with UK Home Office guideline. The following zebrafish lines were used: Wild type WIK and AB, Tg(mbp:EGFP) and EDNRB Hom (Rse tLF802) (*Frohnhöfer et al., 2013*; *Krauss et al., 2014*). Rse zebrafish contain a point mutation in the *ednrba* gene at amino acid 163 where glutamine is substituted for lysine. This mutation creates a restriction site for the enzyme PsiI. Touchdown PCR was performed using a light cycler. To distinguish between wild type and homozygous mutants, the PCR product was incubated with restriction enzyme PsiI for 3 hr at 37°C and run on a 2% agarose TAE gel. Wild type DNA remains uncut at 363 bp whereas homozygous DNA generates two similar sized bands of 187 and 176 bp.

Rse Homozygous mutants were crossed to the stable transgenic line Tg(mbp:EGFP) expressing EGFP in all oligodendrocytes previously generated in the Lyons lab (*Almeida et al., 2011*). Five dpf larval zebrafish were embedded in 1.3% low melting point agarose with tricane. Lateral images of zebrafish spinal cords were taken on an Apatome2 microscope at 20x and stitched together using ZEN imaging software. Using ImageJ cell counter, GFP fluorescent oligodendrocytes were counted along the entire length of the dorsal and ventral spinal cord. Fish were genotyped after imaging and analysis – experimenters were therefore blinded to genotype during the data acquisition.

Offspring of the paired mating of either Rse heterozygotes or Rse heterozygotes to Rse homozygotes were subject to single oligodendrocyte analysis. Single oligodendrocytes were labelled using the mbp:EGFP-CAAX plasmid developed within the Lyons lab (*Almeida et al., 2011*). In brief, plasmid DNA generated above for the mbp:EGFP transgenic line was adapted through the addition of the four amino acid CAAX motif therefore anchoring GFP to membranes of oligodendrocytes and myelin sheaths. Mosaic fish were generated by injecting fertilised eggs with 1 nL of a solution of 12.5 ng/µL mbp:EGFP-CAAX plasmid DNA and 12.5 ng/µL tol2 transposase mRNA between the 1–8 cell stage. Transposase mRNA was produced using an Ambion message machine kit SP6 from pre-digested DNA. Injections were performed using pulled glass needles using gas to force out a precise volume, measured prior to injection in a drop of mineral oil on a calibration slide. four dpf larval zebrafish were screened for fluorescent cells and embedded in 1.3% low melting point agarose with tricane. Lateral images of individual cells were taken on a Zeiss LSM710 confocal. Analysis was performed using ImageJ. GFP fluorescent sheaths longer than 5 µm were traced and the length measured. The number of sheaths made by individual oligodendrocytes was recorded and the mean sheath length was calculated per oligodendrocyte. Fish were genotyped after imaging and analysis - experimenters were therefore blinded to genotype during the data acquisition.

For the experiments asking whether a protein kinase C agonist would rescue myelination defects in the Rse Homozygous mutants, injected fish were treated with either 1%DMSO or 10 µM FR236924 in DMSO at 3 dpf for 24 hr before image acquisition.

## Statistics

All analysis of cell counts and myelination and were performed on ImageJ blind to condition using a filename randomiser macro. Data is presented showing standard deviations to show the variability, or standard error of the mean when averaged data calculated as a mean per animal or per culture was used for each n. Statistical analysis was performed using GraphPad Prism software. Data was tested for normality using a Kolmogorov–Smirnov test. When data fitted a normal distribution parametric t-tests and 1-way ANOVA, with Tukey's post hoc, were used. Where data did not meet normal distribution non-parametric Mann-Whitney U tests and Kruskal-Wallis tests, with Dunns post hoc, were used.

## Acknowledgements

We would like to thank Dr Marie Bechler for help with microfiber cultures and past and present members of the ffrench-Constant, Lyons, Williams and Miron labs for technical assistance and helpful discussions. We thank the University of Edinburgh facilities for animal husbandry and support. This work was supported by a MS Society Research Grant PhD studentship (Grant Reference 950), a Wellcome Trust Senior Investigator Award to CffC and a Wellcome Trust Senior Research Fellowship (102836/Z/13/Z) to DL.

## Additional information

### Funding

| Funder | Grant reference number | Author |
| --- | --- | --- |
| Wellcome | Senior Investigator Award | Charles ffrench-Constant |
| Multiple Sclerosis Society | 950 | Charles ffrench-Constant |
| Wellcome | 102836/Z/13/Z | David A Lyons |

The funders had no role in study design, data collection and interpretation, or the decision to submit the work for publication.

## Author contributions

Matthew Swire, Conceptualization, Data curation, Formal analysis, Investigation, Visualization, Methodology, Writing—original draft, Writing—review and editing; Yuri Kotelevtsev, David J Webb, Resources, Writing—review and editing; David A Lyons, Conceptualization, Supervision, Funding acquisition, Methodology, Writing—review and editing; Charles ffrench-Constant, Conceptualization, Supervision, Funding acquisition, Investigation, Writing—original draft, Project administration, Writing—review and editing

## Author ORCIDs

Matthew Swire (iD) https://orcid.org/0000-0003-4294-4926
David A Lyons (iD) http://orcid.org/0000-0003-1166-4454

## Ethics

Animal experimentation: Animal husbandry and experiments were performed under UK Home Office project licenses issued under the Animals (Scientific Procedures) Act, under project licences 60/8436, 70/8436 and 70/8748. All animal experiments were reviewed, revised and approved by the University of Edinburgh Bioresearch & Veterinary Services team.

## Decision letter and Author response

Decision letter https://doi.org/10.7554/eLife.49493.022
Author response https://doi.org/10.7554/eLife.49493.023

## Additional files

### Supplementary files

• Supplementary file 1. Antibody array of phosphorylation events downstream of EDNRB. Wild type mouse oligodendrocytes were starved for 4–5 hr in media devoid of supplementation and then treated for 15 min with either vehicle or BQ3020 (100 ng/mL). For each antibody the background intensity was subtracted, dye signal normalised and an average calculated of the duplicate spots. The ratio was calculated of binding to the phosphorylated amino acids vs the binding to the non-modified regions of the protein for each molecule, calculating this for both control and BQ3020 treated cells. The fold change in phosphorylation for each targeted amino acid was generated by comparing BQ3020 to vehicle. For selection a fold change of greater than 2 and less than 0.5 was set as the cut-off. Antibody array was performed once – one cell lysate per condition.
DOI: https://doi.org/10.7554/eLife.49493.019

• Transparent reporting form DOI: https://doi.org/10.7554/eLife.49493.020

### Data availability

Data generated from phosphorylation screen is included in Supplementary file 1.

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
