## [Decision Letter]

**Acceptance summary:**

Your work marks an important contribution to the growing field of neuron-glia signaling and the emerging concept of "adaptive" myelination. The identification of signaling molecules and their receptors that inform oligodendrocyte lineage cells about the axonal spiking activity in myelinating fiber tracts has been insufficiently understood, specifically beyond glutamate receptor signaling. The identification of endothelin in this function, has been seen as an attractive solution to the brain’s challenge of regulating both, an energy consuming myelin growth and the necessary supply lines for energy rich metabolites. The implication that with paracrine signaling the response of oligodendrocytes may lose its strict association with single axons is intriguing with respect to the overall role of adaptive myelination.

**Decision letter after peer review:**

Thank you for submitting your article "Endothelin signalling regulates experience-dependent myelination in the CNS" for consideration by *eLife*. Your article has been reviewed by three peer reviewers, including Klaus-Armin Nave as the Reviewing Editor and Reviewer #1, and the evaluation has been overseen by Huda Zoghbi as the Senior Editor. The following individuals involved in review of your submission have agreed to reveal their identity: Julia Edgar (Reviewer #2); Hannelore Ehrenreich (Reviewer #3).

The reviewers have discussed the reviews with one another and the Reviewing Editor has drafted this decision to help you prepare a revised submission.

The paper is interesting and timely for the myelin research field. The data are novel and important and add significantly to our knowledge of how activity modulates myelination. A few specific comments should be addressed in a revision:

1) The authors should discuss the very recent paper by the Monje lab that identified a role for BDNF in adaptive myelination. This factor shares the caveat of endothelin being an "indirect" signaling system that loses axonal specificity but rather serves an entire local area.

2) Discussion: the authors conclude that the "myelination rescuing" effect of intranasally administered PKC agonist in socially deprived mice proves that the underlying defect of social isolation is the reduction of endothelin (i.e. PKC)-mediated stimulation of oligodendrogenesis. We think that is formally incorrect. The intranasal PKC agonist could be a (behaviorally) unrelated promyelinating signal that nevertheless compensates the visible lack of myelin. This argument would be more convincing if the authors could show that (i) also the behavioral consequences of social isolation were rescued by this drug, and (ii) that there is no unspecific hypermyelination beyond the mPFC along the path of diffusion that this drug takes within the CNS. However, we do not say that these experiments must be part of a revision.

3) The authors focus on EDN expressed by endothelial cells/pericytes. They claim that 'in keeping with previous studies, no *Edn1, Edn2* or *Edn3* was observed in S100ß positive astrocytes…' They do not cite the respective source of information for this claim. This is difficult in light of many earlier papers clearly showing astrocytic expression of EDNs using different approaches.

Absence of evidence is not evidence of absence. Has astrocytic expression been checked under any other than baseline conditions? It appears that the contribution of astrocytes to the present findings cannot be excluded. Moreover, in their in situ figure (Figure 2—figure supplement 1) no positive 'dots' are visible in the astrocyte pictures and no dots in the vicinity of astrocytes, i.e. pointing to endothelial cells as an internal positive control that the in situ hybridization has worked. This would make the data more convincing. Only mRNA data on endothelin and EDNRB are shown – are antibody stainings available?

4) For Figures 1H and 3D statistics are not shown, please add (even in case of non-significance).

Generally, please add statistical information to all graphs/figure legends where applicable (e.g. Figure 5C etc.). Also, in some comparisons non-parametric tests were used (e.g. Figure 6D). Please mention in the statistic section on what criteria the decision was based to conduct parametric or non-parametric tests.

5) Figure 3A: The legend mentions group sizes of n=5 and n=4, respectively, for both layers. This is not the number depicted in the graph for layer V; here it is n=9 in WT and n=3 in cKO; what do the dots represent then?

6) Figure 4B: the number of mice used for BQ788 culture is missing. Just for clarity: according to number of cultures, more than one culture was prepared per mouse in control and BQ3020, is this correct? Furthermore, a 1-way ANOVA is mentioned, but for the depicted comparison a T-test would be appropriate (same in Figure 4F).

---

## [Author Response]

1) The authors should discuss the very recent paper by the Monje lab that identified a role for BDNF in adaptive myelination. This factor shares the caveat of endothelin being an "indirect" signaling system that loses axonal specificity but rather serves an entire local area.

We have now included within the discussion the recent work from the Monje lab identifying BDNF-TrkB signalling as a signal influencing adaptive myelination.

“Other pathways could contribute to this aspect of adaptive myelination. Such potential pathways are BDNF signalling via oligodendroglia TrkB (Geraghty et al., 2019, Gibson et al., 2019) and glutaminergic signalling via AMPAR-mediated effects on newly differentiating oligodendrocytes (Kougioumtzidou et al., 2017).”

2) Discussion: the authors conclude that the "myelination rescuing" effect of intranasally administered PKC agonist in socially deprived mice proves that the underlying defect of social isolation is the reduction of endothelin (i.e. PKC)-mediated stimulation of oligodendrogenesis. We think that is formally incorrect. The intranasal PKC agonist could be a (behaviorally) unrelated promyelinating signal that nevertheless compensates the visible lack of myelin.

We agree with the reviewers, and have therefore modified the text to better reflect the

results and state that these “argue strongly” for a role of EDN signalling.

“Our findings showing that social isolation reduces endothelial EDN expression, that an oligodendroglial-specific deletion of the relevant EDNRB receptor phenocopies the myelination defect caused by social isolation, and that an EDNRB agonist rescues the myelination defect of individual oligodendrocytes in the mPFC together argue strongly for a role of vascular EDN synthesis in mediating such indirect effects.”

This argument would be more convincing if the authors could show that (i) also the behavioral consequences of social isolation were rescued by this drug, and (ii) that there is no unspecific hypermyelination beyond the mPFC along the path of diffusion that this drug takes within the CNS. However, we do not say that these experiments must be part of a revision.

We agree that a behavioural rescue would be a powerful result, but one that would be quite surprising given that social isolation likely has effects in addition to those we observe on myelin sheath number. Indeed, we document a reduction in total oligodendrocyte number following isolation, an effect not due to the reduction in EDN signalling and therefore not rescuable by agonist administration. Our conclusion is therefore that the reduction in myelin sheath number resulting from diminished EDN signalling in social isolation likely contributes to the behavioural phenotype, but is not solely responsible for it.

“Our results do not however imply that changes in EDN signalling are solely responsible for the effects of social isolation on myelination. It is clear, for example, that social isolation can affect oligodendrocyte number, a feature that does not appear to be controlled by EDN in the healthy nervous system.”

Regarding the extent of hypermyelination following the intranasal administration of the agonist, further experiments establishing the extent to which EDN signalling can increase or decrease levels of myelination in other regions of the CNS in response to activity are an important next step.

3) The authors focus on EDN expressed by endothelial cells/pericytes. They claim that 'in keeping with previous studies, no Edn1, Edn2 or Edn3 was observed in S100ß positive astrocytes…' They do not cite the respective source of information for this claim. This is difficult in light of many earlier papers clearly showing astrocytic expression of EDNs using different approaches.Absence of evidence is not evidence of absence. Has astrocytic expression been checked under any other than baseline conditions? It appears that the contribution of astrocytes to the present findings cannot be excluded. Moreover, in their in situ figure (Figure 2—figure supplement 1) no positive 'dots' are visible in the astrocyte pictures and no dots in the vicinity of astrocytes, i.e. pointing to endothelial cells as an internal positive control that the in situ hybridization has worked. This would make the data more convincing. Only mRNA data on endothelin and EDNRB are shown – are antibody stainings available?

We have now described and referenced the findings of *Edn1* expression localised to astrocytes following demyelination (Gallo lab). We have also included new figures demonstrating, in the same image, vascular cells positive for *Edn1/Edn3* mRNA expression adjacent to S100β astrocytes or Iba1 microglia that do not express EDN (Figure 2—figure supplement 1). We have been unable to obtain antibodies for EDN or EDNRB that work for immunohistochemistry and therefore can only comment on EDN and EDNRB mRNA transcript expression.

4) For Figures 1H and 3D statistics are not shown, please add (even in case of non-significance). Generally, please add statistical information to all graphs/figure legends where applicable (e.g. Figure 5C etc.). Also, in some comparisons non-parametric tests were used (e.g. Figure 6D). Please mention in the statistic section on what criteria the decision was based to conduct parametric or non-parametric tests.

We have now provided the requested extra statistical information, showing non-significant results and statistical tests used, in relevant figures and legends. We have included information on the rationale behind the parametric and non-parametric tests used throughout in the statistics section.

“All analysis of cell counts and myelination and were performed on ImageJ blind to condition using a filename randomiser macro. […] Where data did not meet normal distribution non-parametric Mann-Whitney U tests and Kruskal-Wallis tests, with Dunns post hoc, were used.”

5) Figure 3A: The legend mentions group sizes of n=5 and n=4, respectively, for both layers. This is not the number depicted in the graph for layer V; here it is n=9 in WT and n=3 in cKO; what do the dots represent then?

We have amended this mistake in the legend to include the correct values: n=9 in WT and n=3 in cKO.

6) Figure 4B: the number of mice used for BQ788 culture is missing. Just for clarity: according to number of cultures, more than one culture was prepared per mouse in control and BQ3020, is this correct? Furthermore, a 1-way ANOVA is mentioned, but for the depicted comparison a T-test would be appropriate (same in Figure 4F).

We have corrected this mistake. For this experiment each culture was prepared from independent mice. Here we have performed a 1-way ANOVA as we are comparing >2 conditions therefore multiple T-tests are less accurate. We are very grateful to the reviewers for spotting this error and the one addressed in 5).